# Update on the Prevalence, Incidence, Mortality, and Trends in Treatment of Inflammatory Bowel Disease in a Population-Based Registry in Catalonia Between 2017 and 2023

**DOI:** 10.3390/jcm14165711

**Published:** 2025-08-12

**Authors:** Eduard Brunet-Mas, Belen Garcia-Sagué, Emili Vela, Caridad Pontes, Luigi Melcarne, Luís E. Frisancho, Laura P. Llovet, Patricia Pedregal-Pascual, Sergio Lario, Maria J. Ramírez-Lázaro, Albert Villoria, Xavier Calvet

**Affiliations:** 1Servei d’Aparell Digestiu, Parc Taulí Hospital Universitari, Institut d’Investigació i Innovació Parc Taulí (I3PT-CERCA), Departament de Medicina, Universitat Autònoma de Barcelona, 08208 Sabadell, Spain; ebrunetm@tauli.cat (E.B.-M.); belengsague@gmail.com (B.G.-S.); lmelcarne@tauli.cat (L.M.); lefrisancho@tauli.cat (L.E.F.); lpllovet@tauli.cat (L.P.L.); slario@tauli.cat (S.L.); mramirezl@tauli.cat (M.J.R.-L.); avilloria@tauli.cat (A.V.); 2CIBERehd, Instituto de Salud Carlos III, 28029 Madrid, Spain; 3Unitat d’Informació i Coneixement, Servei Català de la Salut, Generalitat de Catalunya, 08002 Barcelona, Spain; evela@catsalut.cat; 4Digitalization for the Sustainability of the Healthcare System (DS3), 08002 Barcelona, Spain; 5Servei de Farmacologia Clínica, Hospital de la Santa Creu i Sant Pau, 08025 Barcelona, Spain; cpontesg@gmail.com; 6Departament de Farmacologia, de Terapéutica i de Toxicologia, Universitat Autònoma de Barcelona, 08193 Bellaterra, Spain; 7Servei d’Aparell Digestiu, Hospital de Santa Creu i Sant Pau, Universitat Autònoma de Barcelona, 08025 Barcelona, Spain; patripedregal@gmail.com; 8Departament de Medicina, Universitat Autònoma de Barcelona, 08193 Bellaterra, Spain

**Keywords:** epidemiology, quality of life, socio-economical, psychological endpoints

## Abstract

**Background**: The prevalence of inflammatory bowel disease (IBD) is increasing worldwide, while the incidence is tending to stabilize. Moreover, the use of biological treatments is increasing; some studies suggest that surgeries and hospitalizations are decreasing instead. **Methods**: A population-based, retrospective cohort study was conducted using data from the Catalan Health Surveillance System (CHSS). All patients diagnosed with IBD were included between 2017 and 2023. Crude incidence and prevalence rates were calculated for the Catalan population. Data on pharmacological therapy, surgical procedures, hospitalizations, and mortality were analyzed. Trends in age-sex-adjusted rates were also estimated, and logistic regression was used to calculate the adjusted mortality odds ratio (OR). Data for Crohn’s disease (CD) and ulcerative colitis (UC) were analyzed separately. **Results**: The number of prevalent IBD cases rose from 28,752 in 2017 to 41,423 in 2023. Despite incidence rates remaining stable (30.8 in 2017 and 29.9 per 100,000 inhabitants in 2023), prevalence rates increased (386.9 and 510.9 per 100,000 inhabitants, respectively). The use of biologics significantly increased (from 13.5% in 2017 to 21.0% in 2023), particularly ustekinumab and vedolizumab. In parallel, a decline in the use of immunosuppressants was observed. IBD-related surgeries and hospitalizations decreased during the study period, particularly among CD patients. Mortality remained low but was higher among IBD patients compared to the general population. **Conclusions**: The incidence of IBD in Catalonia has stabilized, while its prevalence continues rising, suggesting a transition to Stage 3 (compounding prevalence). The use of biological treatments is increasing steadily, whereas rates of surgeries and hospitalizations are consistently decreasing.

## 1. Introduction

Inflammatory bowel disease (IBD) comprises two chronic intestinal disorders: Crohn’s disease (CD) and ulcerative colitis (UC) [1]. IBD affects millions of people worldwide and presents a variable clinical course, characterized by alternating periods of remission and flares [2]. Current therapeutic approaches to IBD include the use of therapies such as biological treatments and JAK inhibitors [3,4]. Recent studies have called for the introduction of more aggressive treatment strategies during the early years of the disease; for example, the PROFILE and ACTIVE studies demonstrated higher response rates with combination therapy using infliximab and azathioprine [5,6]. Despite these advances, the reported annual hospitalization rate approaches 20%, and approximately 50% of IBD patients require surgery within 10 years of diagnosis [7,8,9,10].

From an epidemiological perspective, Kaplan et al. characterized four stages of IBD: Stage 1 (emergence) refers to developing regions with low incidence and prevalence; Stage 2 (acceleration in incidence) includes newly industrialized regions with rapidly rising incidence but low prevalence; Stage 3 (compounding prevalence) corresponds to early industrialized regions with steadily climbing prevalence due to the cumulative effect of a much higher incidence than mortality over time; and finally, Stage 4 (prevalence equilibrium) has been hypothesized theoretically as a period in which prevalence plateaus due to the shifting demographics of an aging IBD population, although the existence of this stage has not yet been conclusively demonstrated [11].

In 2011, the Catalan Health Service (CatSalut) set up the Catalan Health Surveillance System (CHSS) database, a population-based health database gathering health information derived from public universal healthcare coverage of the entire population of Catalonia, which in 2023 exceeded eight million inhabitants. Previous studies conducted by our group using this database recorded a high prevalence and incidence of IBD, which both rose between 2011 and 2016, suggesting that Catalonia was in the acceleration in incidence stage (Stage 2) [12]. An increase in the use of biological treatments was also observed, which correlated with a decrease in the rates of surgeries and hospitalizations [13].

The aim of the present study is to provide an update on the current epidemiological status of IBD in Catalonia and to determine whether the region has transitioned to Stage 3 (compounding prevalence). Additionally, we aim to evaluate trends in the use of treatments and their correlation with surgery and hospitalization rates.

## 2. Materials and Methods

### 2.1. Study Design, Participants, and Database

Retrospective analysis of an administrative database that includes all individuals diagnosed with IBD in Catalonia (north-west Europe) between 2017 and 2023 according to the International Classification of Diseases, 10th revision, Clinical Modification codes (ICD-10-CM codes). The ICD-10-CM codes used are shown in Appendix A. A case was considered valid only if an IBD treatment was also recorded.

The CHSS is a comprehensive healthcare information system designed to integrate activity registers from different sources pivoting on the patient, rather than the healthcare provider, thus allowing longitudinal follow-up of care. Since 2018, the CHSS has also included data from hospitals’ outpatient clinics, enhancing both the quantity and quality of its information. The system registers diagnostic information based on a Minimum Basic Data Set aligned with the ICD-CM classification and including data on pharmacy prescriptions, outpatient visits, ambulatory rehabilitation, hospital admissions, and other patient services. It offers a holistic, cross-sectional view of health problems by consolidating data from the entire public health system. Currently, the CHSS serves as a robust, population-based healthcare data source, covering information from the more than eight million residents of Catalonia.

The CHSS includes all diagnoses reported by providers, whether primary or secondary, and employs an automated validation system to ensure data consistency and to detect potential errors. Regular external audits further support its reliability [14].

Data on exposures to different IBD treatments were retrieved from the electronic dispensation records for the same period. IBD treatments included biological drugs and Janus Kinase inhibitors (iJAKs) (infliximab, adalimumab, golimumab, vedolizumab, ustekinumab, tofacitinib, upadacitinib, and filgotinib), immunosuppressive agents (azathioprine, 6-mercaptopurine, and methotrexate), corticosteroids, and salicylates. The list of active principles studied is appended as Appendix A. A patient was considered to have received a particular treatment during a specific year if at least one prescription of the biologic was dispensed during this period.

Data on the number of surgical procedures and the hospitalization rate of IBD patients were also obtained from the CHSS, and both were expressed per 1000 patients/year. ICD-10-CM codes for the different surgical procedures and hospitalizations are detailed in Appendix A.

### 2.2. Statistical Analysis

New cases of IBD were defined as patients residing in Catalonia who were diagnosed with IBD for the first time in each year of the study period (2017–2023). Prevalent cases were defined as the total number of patients with IBD residing in Catalonia and alive on December 31 of each year of the study period and also in relation to the total Catalan population provided by the Institut d’Estadística de Catalunya (IDESCAT) [15], which was used as the reference population for the calculation of incidence and prevalence rates. Data were adjusted by age and sex and expressed per 100,000 inhabitants using Poisson regression models.

The crude mortality rates are expressed per 100 patients and are calculated both by age groups and overall. Causes of death are not explicitly recorded in the database and so could not be determined in this study. Finally, for the year 2023, the probability of dying in the three populations (CD, UC, and no-IBD) adjusted by age, sex, and level of income was calculated by logistic regression. Age was introduced as a categorical variable (15–44, 45–64, 65–74, 75–84, and >84 years) and excluded patients aged below 15 years old. Individual-level income data were obtained from pharmaceutical co-payment groups of the Spanish Social Security System, that are yearly revised and classify insured users based on annual income, categorized as <18,000, 18,000–100,000, and >100,000 euros per year, and also on a number of conditions that qualify for welfare support from the Catalan government. Based on this information, we defined four individual-level socioeconomic status (SES) categories: ‘high SES’ (income >100,000 euros/year), ‘medium SES’ (18,000–100,000 euros/year), ‘low SES’ (<18,000 euros/year), and ‘very low SES’ (individuals receiving government welfare support).

To calculate the annual rate per patient/year for the use of IBD treatments, surgical procedures, and hospitalizations, we estimated the time at risk for each patient in each of the periods. Exposure periods initiated on 1 January (or at the date of diagnosis for incident patients) and ended on 31 December of each year (or with the death of the patient). The sum of the number of users for each treatment or surgical procedure or hospitalization in the period was used as a numerator and the denominator was the number of patients/year at risk; data were expressed in persons per year. The statistical significance of the global variation of rates during this time period was calculated using generalized linear models (Poisson regressions). Pearson correlation coefficients were calculated between the rates of surgical procedures or hospitalizations and the rates of users for each group of drugs. A *p* value of 0.05 or lower was considered significant. The statistical analysis was carried out using the statistical package R, version 4.4.0.

The study was performed and reported according to the STROBE Statement guidelines (http://www.strobe-statement.org/index.php?id=strobe-home (accessed on 15 April 2025)) [16]. The STROBE checklist for reporting studies has been included as Appendix A.

## 3. Results

### 3.1. Patients’ Characteristics

Male/female and age distributions of patients are shown in Figure 1 (see detailed data for CD and UC in Appendix A). The mean age (in 2023) was 50 years for CD and 56.3 years for UC; mean ages increased slightly during the study period (see Appendix A).

### 3.2. Prevalence and Incidence

The number of prevalent IBD cases rose from 28,752 in 2017 to 41,442 in 2023. Of these, approximately two-thirds were UC and the remainder CD. Overall UC-adjusted prevalence in Catalonia was 236.4 per 100,000 inhabitants in 2017 (95%CI 232.9–239.9), increasing to 316.6 in 2023 (95%CI 312.8–320.5). The corresponding figures for CD were 150.5 (95%CI 147.7–153.3) in 2017 and 194.4 (95%CI 191.4–197.4) in 2023 (see detailed data in Table 1). The total number of IBD patients increased notably over the 2017–2023 period. Specifically, there were 8195 more UC patients, a rise of 46.7% in these years, and 4475 more cases of CD, a rise of 25.5%.

Over this period, 19,142 additional incident IBD cases were recorded. The adjusted rates of incidence remained stable over the study period, with an abnormal peak in 2018–2019. UC-adjusted incidence increased slightly from 18.1 per 100,000 inhabitants in 2017 (95%CI 17.5–19.4) to 19.6 in 2023 (95%CI 18.7–20.6). As for CD, the incidence decreased from 12.4 (95%CI 11.6–13.2) in 2017 to 10.3 (95%CI 9.6–11) in 2023 (see detailed data in Table 1 and Figure 2).

### 3.3. Mortality

Mortality rates presented small variations over the course of the study period; the mean mortality rate was 1.5 per 100 inhabitants, and the maximum rate was 1.7 per 100 (Table 1). Figure 3a shows the 2023 mortality rates broken down by age. The mortality rate was very low until the fifth decade of life, and then increased gradually. When compared with the non-IBD population, an excess mortality of IBD patients was observed. The age- and sex-adjusted odds ratios (OR) of death of IBD patients in 2023 were significantly higher both for patients with CD (OR: 1.580; 95%CI: 1.362–1.833) and for patients with UC (OR: 1.213; 95%CI: 1.097–1.342) (Figure 3b).

### 3.4. Time Trends in the Use of Drugs

Salicylates were used in half of the UC patients (50.8% in 2023), but in CD, the number of users was much lower, falling over the study period from 16.1% in 2017 to 13.2% in 2023 (*p* < 0.001 for both CD and UC).

The number of users of immunosuppressive treatment also fell in both UC and CD patients, from 12.3% and 36.8% in 2017 to 11% and 28%, respectively, in 2023. The difference was statistically significant for CD (*p* < 0.001) but not for UC (*p* = 0.146).

The use of biological treatments increased from 13.5% in 2017 to 21% of users in 2023 (*p* < 0.001 for both CD and UC) (Appendix A).

With regard to the use of biological therapies and iJAKs (Appendix A), infliximab was the most commonly administered in 2017, being applied to 1807 patients (6.3% of all IBD cases and 43.6% of all patients receiving biologics). This number increased to 2865 in 2023; although the percentage of IBD users remained relatively stable (6.9% of all IBD patients), the percentage of those receiving biologics decreased (30.8%). Adalimumab use rose significantly, from 1795 patients in 2017 (6.2% of all IBD patients and 43.2% of patients receiving biologics) to 3281 in 2023 (7.9% of all IBD patients and 34.8% of patients receiving biologics), making it the most widely used treatment in 2023. The use of golimumab was limited, peaking at 176 patients in 2019 (0.5% of all IBD patients and 3% of patients receiving biologics).

The administration of ustekinumab increased markedly, from 133 patients in 2017 (0.5% of all IBD patients and 3.2% of patients receiving biologics) to 2001 in 2023 (4.8% of all IBD patients and 20% of patients receiving biologics), establishing it as the second most used treatment after anti-TNF agents. Vedolizumab use also rose, from 275 patients (1% of all IBD patients and 6.6% of patients receiving biologics) in 2017 to 831 (2% of all IBD patients and 9% of patients receiving biologics) in 2023.

Among JAK inhibitors, tofacitinib was the most commonly used, rising from just two patients in 2017 (0.01% of all IBD patients and 0.4% of patients receiving biologics) to 185 in 2023 (0.4% of all IBD patients and 1.9% of patients receiving biologics). Upadacitinib and filgotinib were used in only a few cases, both off-label during the study period.

The trend in the use of all biologics was similar in CD and UC, except for golimumab, which was predominantly used in UC, and adalimumab, which was primarily used in CD.

### 3.5. Surgical Procedures

The absolute number of ostomies and resections rose slightly from 2017 to 2023: ostomies from 204 to 233, and resections from 288 to 360 (Appendix A). However, as the prevalence of IBD during this period of time increased markedly, the rate per 1000 patients/year decreased significantly in both cases: ostomies from 7.4 to 5.8 (*p* = 0.015) and resections from 10.4 to 9.0 (*p* = 0.026) (Appendix A).

Patients with CD underwent surgery almost twice as often as those with UC. In 2023, ostomy rates in these two conditions were 7.6% and 4.7%, respectively, and resection rates reached 11.2% and 7.6%.

Overall, for IBD, a significant positive correlation was observed between the use of immunosuppressants and the need for ostomies (r = 0.83; *p* < 0.05). On the other hand, biological treatment was negatively correlated with the need for ostomies (r = −0.79; *p* < 0.05). Specifically, for CD, the use of salicylates and immunosuppressants demonstrated a significant positive correlation with both surgical procedures, ostomies (r = 0.83; *p* < 0.05 and r = 0.78; *p* < 0.05, respectively) and resections (r = 0.78; *p* < 0.05 and r = 0.77; *p* < 0.05, respectively). No other significant correlations were found (Appendix A).

### 3.6. Hospital Admissions

The number of IBD patients requiring hospitalization for any cause rose from 7147 in 2017 to 10,121 in 2023; the rates per 1000 inhabitants remained stable (258.8 and 252.5, respectively).

Regarding the specific causes of hospitalization, IBD-related hospitalization fell from 34.4 per 1000 patients in 2017 to 27.3 per 1000 patients in 2023 (*p* < 0.001). Patients with CD were more than twice as likely to be hospitalized as those with UC. Similarly, infection-related hospitalizations also fell from 10.2 to 8.9 per 1000 patients (*p* = 0.06). Detailed data on other causes of hospitalization are provided in Appendix A.

Overall, for IBD, a positive correlation was found between the use of corticosteroids and the need for hospitalization for any cause (0.77). The use of immunosuppressants also showed a positive correlation with the need for hospitalization due to IBD (0.87). On the other hand, biological treatment presented a negative correlation with the need for hospitalization due to IBD (−0.86). Specifically, for CD, biological treatment negatively correlated with the need for hospitalization due to IBD (−0.87), while for UC, the use of corticosteroids was positively correlated with hospitalizations due to any cause (0.91). No other significant correlations were found (Appendix A).

## 4. Discussion

This study presents an update on the epidemiology of IBD in Catalonia.

Between 2011 and 2016, increases in both the prevalence and incidence of IBD were observed, suggesting that Catalonia was in Stage 2 (acceleration of incidence) [12]. In the present study, incidence rates were shown to have stabilized, while prevalence continued its upward trend, suggesting a transition to Stage 3 (compounding prevalence).

These trends are in line with observations from other regions currently transitioning to Stage 3. In Olmsted County, MN, USA, the prevalence of CD and UC increased from 174 and 214 cases per 100,000 persons in 2001 to 246.7 and 286.3 cases per 100,000 persons, respectively, in 2011. The incidence increased steadily between 1970 and 2010; however, in recent years, it has shown a tendency to stabilize and plateau [17]. Similarly, in Denmark, the prevalence of IBD increased significantly from 58 cases per 100,000 person-years in 1980 to 893 per 100,000 person-years in 2017. During the same period, however, the incidence of CD and UC rose from 5.1 and 6.2 to 15.6 and 27.2 per 100,000 person-years, respectively [18].

Other countries seem to have already transitioned to Stage 3. In Scotland, the prevalence of IBD increased from 567 per 100,000 inhabitants in 2008 to 784 per 100,000 inhabitants in 2017. Meanwhile, the incidence of IBD stabilized between 2012 and 2017, fluctuating between 42.9 and 40.8 per 100,000 inhabitants [19]. In Veszprem, Hungary, the prevalence of CD and UC in 2011 was 191.18 and 283.51 per 100,000 inhabitants, rising to 236.78 and 317.79 per 100,000 inhabitants, respectively, by the end of 2015. The incidence rates of CD and UC during this 12-year period initially increased, but later plateaued and decreased slightly, with mean incidence rates of 9.9 and 11 per 100,000 person-years, respectively [20,21]. In Israel, there was a significant increase in the prevalence of CD and UC, from 10,383 and 10,737 patients in 2005 to 24,934 and 20,875, respectively, in 2017. However, the overall incidence of IBD fell slightly, from 31.3 per 100,000 inhabitants in 2005 to 25.4 per 100,000 inhabitants in 2017 [22].

We have observed a notable shift in the age distribution of patients with IBD. In 2023 the proportional representation of the population in older age groups (above 60 years) was higher than during the 2011–2016 period [12]. The reason may have been that, although IBD is typically diagnosed earlier in the lifespan, therapeutic advances, improvements in the control of this disease, and the low related mortality contributed to the aging of the IBD population [23]. This progressive aging in the IBD population has been observed in some studies [24,25,26] and is forecast in others [19].

As far as treatments are concerned, the use of biological treatments presented a huge increase, while the use of classical immunosuppressants fell notably. This could be explained by the increasing availability of biological therapies and new drugs used, usually, in monotherapy, and the aging of the IBD population just mentioned, which may have contraindications and increased risk for adverse events with non-biologic immunosuppressant treatment.

As previously reported [13], this study also demonstrated that the increase in the use of biological treatments may lead to a reduction in surgery and hospitalization rates, especially in IBD-related hospitalizations. We stress that this reduction is achieved without an increase in hospitalizations for infections or neoplasms. The fact that our group observed similar findings over time reinforces the internal consistency of our study, suggesting that the results are not isolated or incidental [12]. Several studies also support these findings. In a national study in Israel, Atia et al. observed that the increase in the use of biological treatments from 8.9% in 2005 to 36% in 2019 was accompanied by falls in the hospitalization and surgery rates. Additionally, the median time to hospitalization and surgery also decreased, from 1.3 and 4.7 years in 2005 to 0.2 and 0.6 years in 2019, respectively [27]. Khoudari et al. reported data from a commercial database within the US healthcare system, finding that IBD patients treated with biologics were significantly less likely to undergo bowel resection (9.2%) than those who had never received these agents (12.1%) [28]. However, some studies do not fully align with these results. In a study conducted in Ontario between 1995 and 2012, Murthy et al. reported that the use of anti-TNF treatment did not reduce CD-related hospitalizations (OR 1.06, 95%CI: 0.81–1.39) or intestinal resections (OR 1.10, 95%CI: 0.81–1.50). Similarly, no reductions were observed in UC-related hospitalizations (OR 1.22, 95%CI: 1.07–1.39) or colectomies (OR 0.93, 95%CI: 0.54–1.61) [29]. Other studies are in partial agreement. Fansiwala et al. reported that, although the overall rate of surgeries fell with the increased use of biological treatments, surgeries for bowel obstruction rose slightly. They theorized that the effect of medical therapy may have a limited impact, particularly in cases of stricturing disease [30].

Regarding mortality, this study shows that IBD patients have a slight increase in the relative risk of mortality compared to the non-IBD population, similar to those observed in previous studies [12]. This small increase achieves significance due to the large number of individuals included in the study. Unfortunately, we were unable to determine the causes due to the methodology of the register.

This study has certain limitations. First, the comparison with the previous study must be interpreted cautiously. During the ensuing period, a new version of the ICD-CM codes was introduced (version 10 in place of version 9), and the CHSS was updated, significantly improving the identification of IBD cases by applying stricter criteria and excluding doubtful cases. As a result, while it is possible to compare dynamics in prevalence, incidence, and rates, direct comparisons of absolute numbers or exact rates per inhabitant are not appropriate.

Second, the CHSS database includes only patients who use the public health system. Although it does not capture data from private healthcare due to differing patient identification codes, the impact of this shortcoming is expected to be minimal, because of the wide use of universal public coverage by the population, as the high costs of treating chronic conditions like IBD typically require patients to access publicly funded care. Additionally, as receiving IBD treatment is a condition for identification of IBD patients, some patients with mild IBD who have never received specific IBD treatment may not be recorded.

Furthermore, as IBD is a chronic disease and pharmacological treatments are relatively costly, few private insurance policies cover expensive biologic treatments and most patients who require these drugs are controlled in the public system, which funds 100% of the biological treatment cost. Thus, the dispensing of biologics can be assumed to be almost 100% public. Additionally, another potential limitation of this study is that immigrant patients who arrive in Catalonia with prior IBD treatment are considered incident cases when they first come into contact with the public healthcare system; this may result in a slight overestimation of the true incidence. As a final limitation, as in every epidemiological study, causality cannot be proved for the correlations and trends observed.

As stated above, the prevalence of IBD in Catalonia has been rising since 2011. Although incidence seems to have stabilized, the prevalence of IBD will continue to rise for many years. This highlights the urgent need to prepare our healthcare system to manage this growing patient population.

Additionally, as the IBD population ages, the presence of comorbidities will increase, making patient management more complex and requiring the application of a more holistic approach. It is thus essential to ensure appropriate medical, nursing, and pharmacy resources, as many of these patients will require expert management and complex therapies and support. Significant adaptations in the current healthcare system will be essential to manage the increasing prevalence of IBD and the aging population in order to ensure sustainable and resilient long-term care.

In conclusion, our study demonstrates that the incidence of IBD in Catalonia has stabilized, while its prevalence continues to rise, suggesting a transition to Stage 3 (compounding prevalence). The study also shows that IBD patients have a small increase in mortality compared to the non-IBD population. The use of biological treatments is increasing steadily, whereas rates of surgeries and hospitalizations are consistently decreasing.

## Figures and Tables

**Figure 1 jcm-14-05711-f001:**
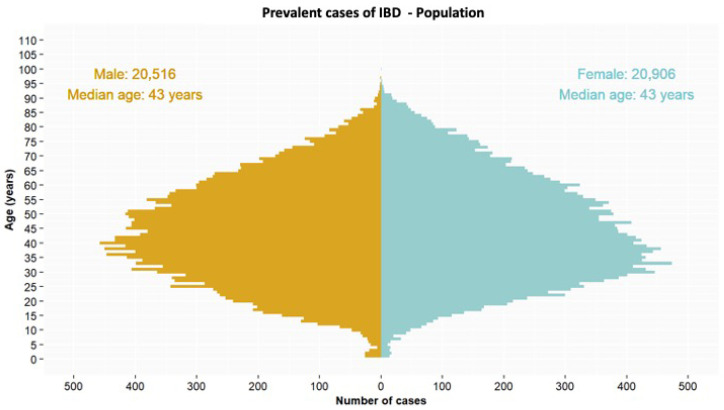
Sex and age distribution for IBD patients in Catalonia in 2023.

**Figure 2 jcm-14-05711-f002:**
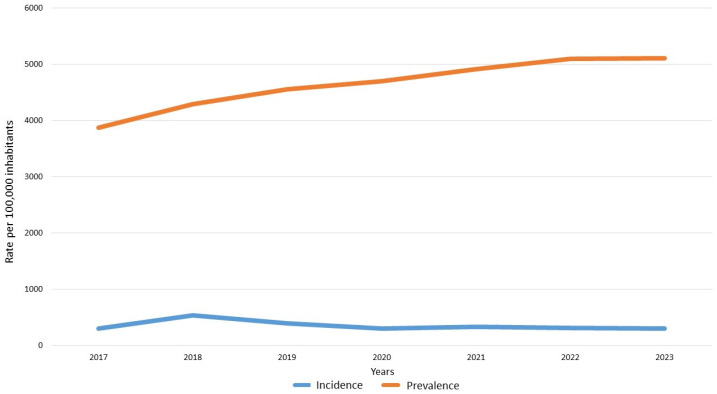
Incidence and prevalence of IBD in Catalonia between 2017 and 2023.

**Figure 3 jcm-14-05711-f003:**
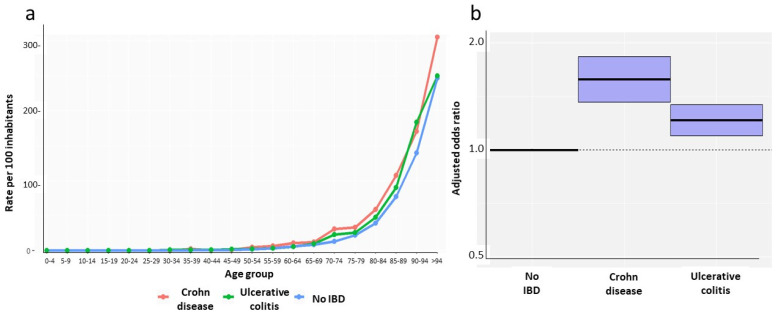
Mortality rates of IBD patients by age (**a**) and age- and sex-adjusted odds ratios of death due to IBD in Catalonia in 2023 (**b**).

**Table 1 jcm-14-05711-t001:** Incidence, prevalence, and mortality for UC, CD, and IBD between 2017 and 2023. Incidence and prevalence are expressed per 100,000 inhabitants and mortality per 100 inhabitants.

		2017			2018			2019			2020			2021			2022			2023		
		n	Rate	IC95%	n	Rate	IC95%	n	Rate	IC95%	n	Rate	IC95%	n	Rate	IC95%	n	Rate	IC95%	n	Rate	IC95%
UC	Incidence	1.373	18.1	17.5– 19.4	2597	34.6	33.3– 35.9	1906	25.1	24.0– 26.2	1547	20.1	19.1– 21.1	1735	22.4	21.4– 23.5	1637	21.0	20.0– 22.0	1589	19.6	18.7– 20.6
	Prevalence	17,533	236.4	232.9– 239.9	19,746	263.6	260.0– 267.3	21,261	280.2	276.4– 284.0	22,300	289.3	285.5– 293.1	23,531	303.4	299.6– 307.3	24,644	315.4	311.5– 319.4	25,728	316.6	312.8– 320.5
	Mortality	272	1.5		307	1.5		325	1.5		449	2		423	1.8		453	1.8		419	1.6	
CD	Incidence	925	12.4	11.6– 13.2	1421	18.9	17.9– 19.9	1076	14.1	13.3– 15.0	824	10.7	10.0– 11.4	848	11.0	10.2– 11.7	839	10.8	10.1– 11.6	825	10.3	9.6– 11.0
	Prevalence	11,219	150.5	147.7– 153.3	12,426	165.3	162.4– 168.2	13,304	175.0	172.0– 178.0	13,880	179.9	177.0– 182.9	14,527	187.5	184.5– 190.6	15,133	194.4	191.3– 197.5	15,694	194.4	191.4– 197.4
	Mortality	112	1.0		160	1.3		148	1.1		194	1.4		153	1		195	1.3		194	1.2	
IBD	Incidence	2298	30.8	29.6– 32.1	4018	53.5	51.8– 55.1	2982	39.2	37.8– 40.7	2371	30.7	29.5– 32.0	2583	33.3	32.1– 34.7	2476	31.8	30.6– 33.1	2414	29.9	28.7– 31.1
	Prevalence	28,752	386.9	382.5– 391.4	32,172	428.9	424.3– 433.7	34,565	455.2	450.4– 460.0	36,180	469.2	464.4– 474.1	38,058	491.0	486.0– 495.9	39,777	509.8	504.8– 514.8	41,422	510.9	506.0– 515.9
	Mortality	384	1.3		467	1.4		473	1.3		643	1.7		576	1.5		648	1.6		613	1.5	
Population of Catalonia		7,496,276			7,543,825			7,619,494			7,722,203			7,739,758			7,758,615			8,016,606		

## Data Availability

For ethical/privacy reasons, the data cannot be shared publicly because they come from a population registry supervised by the Catalan Health Surveillance System and can only be accessed from CatSalut. The data will be shared upon reasonable request made to the corresponding author.

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
