# Peer review of "Update on the Prevalence, Incidence, Mortality, and Trends in Treatment of Inflammatory Bowel Disease in a Population-Based Registry in Catalonia Between 2017 and 2023"

_jcm, 2025, doi:10.3390/jcm14165711_

Round 1
Reviewer 1 Report
Comments and Suggestions for Authors
This article provides valuable data on the epidemiology of inflammatory bowel diseases in Catalonia, offering regional insights from Spain during a period marked by expanding use of biologics. The study uses robust real-world data from a comprehensive population-based health database, presenting clinically relevant trends in incidence, prevalence, treatment patterns, and outcomes.
Comments on the Quality of English LanguageThe manuscript would benefit from minor grammatical corrections to improve clarity and readability. Two examples include:
- Page 6 of 12
Original: “This number increased to 2,865 in 2023, although the percentage of IBD users remained relatively stable (6.9% of all IBD patients) they percentage of those receiving biologics decreased (30.8%).”
Correction: Change "they" to "the". - Page 7 of 12
Original: “During the 2011 and 2016 increases were in both the prevalence and incidence of IBD observed, suggesting that Catalonia was in IBD Stage 2 (acceleration of incidence).”
Correction: Revise for clarity. Suggested:
“Between 2011 and 2016, increases in both the prevalence and incidence of IBD were observed, suggesting that Catalonia was in Stage 2 (acceleration of incidence).”
Author Response
Reviewer 1.
- Page 6 of 12
Original: “This number increased to 2,865 in 2023, although the percentage of IBD users remained relatively stable (6.9% of all IBD patients) they percentage of those receiving biologics decreased (30.8%).”
Correction: Change "they" to "the".
Thank you for the positive comments and for detecting those typo errors. We have already corrected it and checked again the manuscript for additional mistakes.
- Page 7 of 12
Original: “During the 2011 and 2016 increases were in both the prevalence and incidence of IBD observed, suggesting that Catalonia was in IBD Stage 2 (acceleration of incidence).”
Correction: Revise for clarity. Suggested:
“Between 2011 and 2016, increases in both the prevalence and incidence of IBD were observed, suggesting that Catalonia was in Stage 2 (acceleration of incidence).”
Thank you again for detecting the typo error.
Reviewer 2 Report
Comments and Suggestions for Authors
Comparisons of data from 2017 and 2023 in the same region show a clear transition from Stage 2 to Stage 3 according to Kaplan et al.'s classification.
One concern is that in Figure 3a, the disease-related mortality rate appears to be the same as that in the No IBD group, but in Figure 3b, the adjusted odds ratio for disease-related mortality is higher in the IBD group. How can this difference be explained? Generally, the life expectancy of IBD patients is considered to be the same as that of the No IBD group. Is there a difference in life expectancy in this cohort?
Author Response
Comparisons of data from 2017 and 2023 in the same region show a clear transition from Stage 2 to Stage 3 according to Kaplan et al.'s classification.
One concern is that in Figure 3a, the disease-related mortality rate appears to be the same as that in the No IBD group, but in Figure 3b, the adjusted odds ratio for disease-related mortality is higher in the IBD group. How can this difference be explained? Generally, the life expectancy of IBD patients is considered to be the same as that of the No IBD group. Is there a difference in life expectancy in this cohort?
Thank you for the positive comments. The discrepancy detected by the reviewer is only apparent. The increase in mortality is minimal and curves only diverge after the age of 65. The OR, in consequence, are very low, below 1,5. The differences are significant due to the extremely high population included in the study. Despite this finding, this study cannot allow to analyses causalities. We have included this point in the discussion.
Reviewer 3 Report
Comments and Suggestions for Authors
Dear Erudite Editor of the Journal of Clinical Medicine and Esteemed Authors of the manuscript jcm-3794834, thank you for inviting me to review this submission. Population-based retrospective studies are essential to advancing epidemiological information in a country or relevant territory. Therefore, I believe that this manuscript has the potential to be published in this journal. I think the comments below will help the authors improve their manuscript sufficiently before the final decision. The most crucial flaw is your methodology section. To ensure methodological rigor and transparency, the following components should be clearly described when conducting and reporting a population-based, retrospective cohort study. Your current methodology is poorly defined and does not reflect the current state of methodological rigor, which is necessary to publish a well-designed manuscript.
- Therefore, the study design must be explicitly stated, specifying that it is a retrospective cohort study. This designation should be mentioned in the title, abstract, and methods section. Next, the setting should be described in detail, including the source population from which the cohort is drawn. This involves specifying the type of data source used, such as a national registry, health insurance database, or hospital system, as well as outlining the geographic region and the time covered by the study. Please include the necessary information regarding registration protocol in government datasets, or even the protocol generated by each search you’ve done in these datasets, if available. Please include the keywords used in the searches, along with the relevant results based on these keywords. Include if you used Boolean Operators, as well as if you used filters to refine the search of your manuscript. Please include this information with additional details. You can also use a table to ensure that all relevant information is sufficiently gathered. This information must be collected in dedicated subsections to ensure the reproducibility of your findings.
- The participants and cohort definition require explicit inclusion and exclusion criteria. It is essential to specify how individuals were identified within the dataset, the method of cohort assembly, and the approach used to follow individuals over time. The dates marking the start of cohort entry and the end of follow-up should be explicitly stated. Exposure should be clearly defined by describing what constitutes exposure, how it was measured, and the timing of exposure assessment relative to follow-up. I believe that this additional content would undoubtedly enhance your manuscript’s quality and readability. I think that you should clearly define your inclusion and exclusion criteria, alongside the other information mentioned above, with additional detail in your manuscript to ensure rigor and consistency of the population-based survey.
- Outcomes must be more precisely specified, including primary and secondary outcomes, with clear definitions and descriptions of how they were measured or ascertained, for example, using diagnostic codes or laboratory values. In addition, these should be interpreted in terms of relevance. Therefore, include the relevance behind evaluating each of these factors. You should use references to reference your points of discussion. Covariates and potential confounding variables should also be listed with justification for their inclusion based on subject-matter expertise or previous literature with excessive detail. The data sources and measurement section should describe the databases or data systems used, such as electronic medical records, claims data, or administrative health databases, along with any available information on the validity and reliability of the data. This would undoubtedly enhance your manuscript’s exploration in detail. Therefore, you should gather the information mentioned above in your manuscript’s methodology using subsection approaches.
- Potential sources of bias, including selection bias, information bias, or misclassification, should be identified, and the strategies used to mitigate these biases must be described. Although retrospective, the sample size and power should be discussed with a rationale for the adequacy of the study size to detect meaningful associations. Finally, the statistical methods should be more comprehensively detailed, including the types of models used, such as Cox proportional hazards or logistic regression, the handling of missing data, and the conduct of sensitivity analyses. You should use references to reference these points in your discussion. If subgroup or stratified analyses are performed, their rationale and methods should be clearly outlined. You should also use references to reference your points of statistical significance discussion. This would undoubtedly enhance the quality and clarity of your manuscript’s analyses and improve the reporting of your findings. Your statistical analyses section is poorly described as it is now. Therefore, these modifications are of utmost necessity, since population-based studies are fragile.
- Your results section could also be improved to ensure rigor and transparency in reporting your findings. The cohort description should include a more straightforward presentation of the study population, often illustrated by a flow diagram that shows the number of individuals at each stage of the study, from initial eligibility through inclusion and follow-up. Baseline characteristics of the exposed and unexposed groups should be described in detail to allow for a clear comparison of the groups at the start of the study. You can use a prognostic and demographic table to present these findings with appropriate rigor and consistency. The main results should provide effect estimates such as relative risks or hazard ratios, accompanied by confidence intervals and p-values to convey the precision and statistical significance of the findings. Note that some of your results do not possess sufficient description of the respective p-values, which is undoubtedly an error. Additional analyses should include any subgroup analyses that were conducted with further detail of content to explore whether associations varied within specific segments of the cohort. Sensitivity analyses or secondary analyses should also be reported with further detail of content to assess the robustness of the primary findings and to investigate alternative explanations or potential biases. Your results should prioritize the data above to be presented in your manuscript. The current results section of your text does not clearly state all the necessary information regarding the potential of your survey-based analysis. You should avoid errors and report your results consistently.
- The discussion of your manuscript should also be improved based on the modifications above. The debate should begin with a clearer summary of the key findings, highlighting the main results of the study. Following this, the findings should be put into context by comparing them with those of previous studies, explaining similarities or differences, and contributing to the existing body of knowledge. The way it is now, your discussion does not sufficiently describe the current state of the debate, which is necessary for an in-depth exploration of the results of your present population-based article. It is essential to also discuss with additional detail the strengths and limitations of the present study, addressing aspects such as the study design, generalizability of the results, potential sources of bias, and the possibility of residual confounding. The discussion should also explore the implications of the findings for clinical practice, public health, or policy, emphasizing their relevance and potential impact. Finally, the section should conclude with a concise statement of the main take-home message, summarizing what the reader should remember from the study. This should include future research directions based on your findings. An illustration built using BioRender or Mind the Graph should also be included to depict the current state of knowledge based on your findings and to improve the necessary information based on the modifications you’ve made in your discussion section.
- Your manuscript completely lacks a conclusion section. Please include a conclusion section based on the previous sections of your manuscript that clearly states the current state of knowledge.
- Your supplementary material is currently disorganized. Please organize the supplementary material of your manuscript by ensuring that all relevant material is depicted in the sections of the manuscript.
- Please ensure the rigor and formatting while assessing and reporting your findings. Respect paragraphs and bullet points where applicable, as well as the correct design of section and subsection descriptions.
- Your manuscript would gather more citations if you could include a dedicated scientific illustration delving into the IBD physiopathology in detail in your introduction section. Also, including a graphic abstract is highly recommended.
- Your manuscript has a high similarity index. This is unacceptable. The authors must lower this index immediately.
Thank you for your attention to this matter.
I look forward to receiving a revised version of this manuscript shortly.
With best regards and always available,
The Reviewer.
Author Response
The response is available in the attachment.

Round 2
Reviewer 3 Report
Comments and Suggestions for Authors
Thank you for your response.